# ORFLEX: Orthogonal Reparameterization with Flexibility for Multimodal Large Language Model Fine-Tuning

## Abstract

Parameter-Efficient Fine-Tuning (PEFT) has emerged as a key strategy for adapting pretrained large models with minimal trainable parameters. While most methods were developed for LLMs and later extended to multimodal domains, their direct application to multimodal large language models (MLLMs) often overlooks modality-specific discrepancies. In particular, although visual tokens are aligned with language tokens in feature space, differences persist during forward propagation, which existing LoRA-based approaches fail to address. In this work, we propose ORFLEX, a reparameterized PEFT method tailored for MLLMs. First, we observed that the LoRA column spaces associated with visual and text tokens tend to be strongly orthogonal when the parameters are decoupled. Further, we leveraged this property by introducing modality-specific reparameterization branches and designing a QR-inspired decomposition of the LoRA matrix into a frozen orthogonal basis $\hat{Q}$ and a lightweight learnable matrix $\hat{R}$. In addition, we incorporated learnable Householder transformations to adaptively rotate $\hat{Q}$ while preserving orthogonality, enhancing expressiveness. Extensive experiments demonstrate that our approach consistently outperforms strong baselines on both general and domain-specific multimodal benchmarks, underscoring the effectiveness of modality-aware reparameterization to advance PEFT for MLLMs.

## 1 Introduction

Parameter-Efficient Fine-Tuning (PEFT) has emerged as a crucial technique for adapting pretrained large models (PLMs) . Compared to full-parameter fine-tuning, PEFT requires fewer trainable parameters, enabling model optimization under limited GPU memory constraints (Han et al., 2024).

In recent years, researchers have extensively explored efficient fine-tuning strategies. For example, soft prompt methods such as prefix-tuning (Li & Liang, 2021) and prompt-tuning (Lester et al., 2021) introduce learnable dense tokens while keeping the backbone frozen. Selective tuning methods like BitFit (Zaken et al., 2022) and FAR (Vucetic et al., 2022) update only a small subset of model parameters. Among these methods, reparameterization approaches such as LoRA (Hu et al., 2022) introduce low-rank adaptation to the original weight matrices rather than directly modifying them, which have remained one of the most widely used PEFT techniques.

Many PEFT methods were initially developed for LLMs and have since been extended to the computer vision domain. Several PEFT methods have also been developed specifically for visual or multimodal models, such as those targeting diffusion models (Farhadzadeh et al., 2025; Xia et al., 2025) and vision-language models (Ghiasvand et al., 2025; Wang et al., 2025). But for multimodal large language models (MLLMs), due to their architectural similarity to LLMs' backbones, it is often assumed that PEFT techniques designed for LLMs can be directly applied. However, some evidence indicates that this approach may not be well justified. As shown in Figure 2, we extracted visual and text embeddings from image-text pairs in two datasets and applied UMAP for dimensionality reduction. The results reveal a clear separation between the two types of embeddings. Additionally, as noted in previous works (Chen et al., 2024; Zhang et al., 2024), the attention score matrices in MLLMs' intermediate layers also show distinct boundaries between visual and text token regions. These findings suggest that, despite aligning visual tokens with language token feature spaces, dis-

crepancies between modalities persist during the forward pass, which current LoRA-based methods overlook. Thus, there is significant potential to improve PEFT methods for MLLMs.

Building on these insights, we conduct an in-depth analysis of the unique behaviors arising from the differences between image and text tokens, and based on these findings, we propose a reparameterized PEFT method tailored for MLLMs. Specifically, unlike existing approaches that typically manipulate the original LoRA matrices through singular value decomposition (Meng et al., 2024; Zhang et al., 2023; Fan et al., 2025) or gradient analysis (He et al., 2023; Wang et al.), we introduce separate reparameterization matrices for visual and language tokens to decouple their parameter spaces. Our subsequent analysis reveals that the column spaces of the two LoRA $A$ matrices tend to be more orthogonal, and since the column space of $A$ reflects the token projection space during the forward pass, this orthogonality becomes a critical property.

To leverage this and reduce covariance in tokens during forward propagation, we design two parallel reparameterization branches, and inspired by QR decomposition, we further decompose each $A$ matrix into a frozen orthogonal matrix $\hat{Q}$ and a learnable module $\hat{R}$. This design allows us to maintain the orthogonality between the two $A$ matrices, while enabling active expansion of the column space by increasing the rank of $\hat{Q}$ at minimal cost.

Moreover, to further enhance the adaptability of the orthogonal basis in $\hat{Q}$, we incorporate learnable Householder transformations. This allows the column space of $\hat{Q}$ to undergo adaptive transformations, such as global rotations while preserving mutual orthogonality, thereby improving the overall expressive capacity of the model.

In general, the contributions of our work can be summarized as follows:

- Based on the fact that modality tokens differ in feature space, we conducted decoupling analysis of PEFT parameters, and made a new discovery that column spaces of parameter matrices associated with different modalities exhibit strong orthogonality.
- Building on this discovery, we designed a novel PEFT method for MLLMs based on matrix theories like QR decomposition and Householder transformation, achieving orthogonality in the column spaces of modality-specific matrices while maintaining high adaptability.
- Our method, specifically designed for MLLMs, achieves state-of-the-art performance on both general-purpose and domain-specific multimodal benchmarks.

## 2 RELATED WORKS

**Reparameterization-based PEFT methods.** PEFT methods, exemplified by LoRA (Hu et al., 2022), achieve efficient training by reparameterizing weight matrices with few learnable parameters. Subsequent work has explored several main directions. Some methods leverage Singular Value Decomposition (SVD) to identify and update dominant components (Meng et al., 2024; Zhang et al., 2023; Fan et al., 2025). And another category reduces redundancy in the reparameterization matrices to save parameters and speed up training (Li et al., 2024; Kopiczko et al., 2024). A third line of work analyzes the parameter gradients, aiming to guide updates in a more optimal direction (He et al., 2023; Wang et al.). In contrast, our method takes a different perspective by viewing matrix multiplication as a projection and focusing on the matrix subspace.

**PEFT for vision models.** In recent years, researchers have proposed several PEFT methods specifically tailored for vision-related models. For diffusion models, techniques such as Kronecker products and quantization have been introduced (Liu et al., 2025b; Marjit et al.; Guo et al., 2024). For vision-language models similar to CLIP (Radford et al., 2021), methods like feature adaptation and instruction enhancement have been explored to enable efficient fine-tuning (Hegde et al., 2025; Zhai et al., 2023; Xu et al., 2025). However, for MLLMs, existing work tends to directly apply general-purpose PEFT methods. For example, DoRA (Liu et al., 2024) has been evaluated in MLLM with multimodal tasks. To the best of our knowledge, no prior work has conducted a systematic analysis or developed fine-grained designs for PEFT in MLLM. Our work fills this gap.

**The subspace of weight matrix.** In forward propagation, the parameter matrix subspace plays a crucial role. Several studies have explored this and revealed useful insights. Garg et al. (2024) shows that models often underuse the rank of parameter matrices. Hwang et al. (2025) improves efficiency

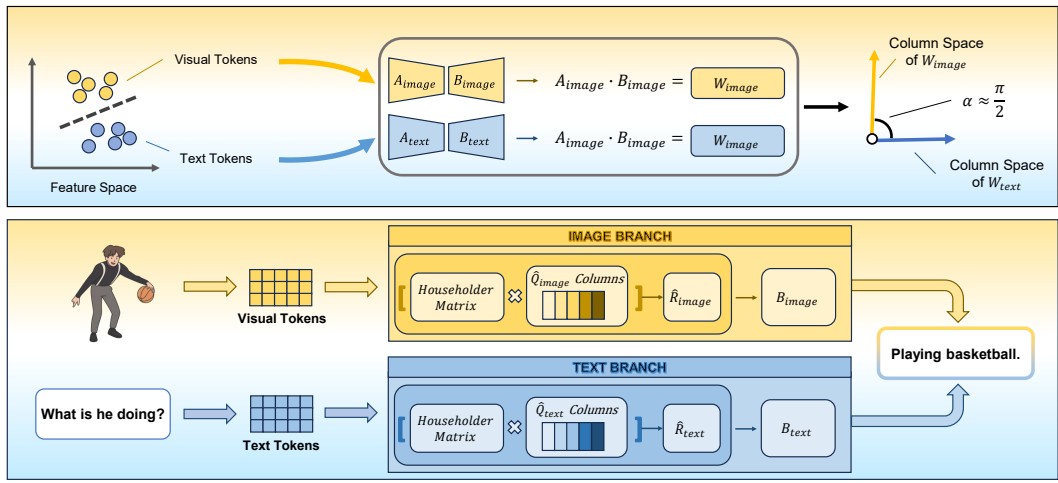

Figure 1: The upper part of the figure shows our parameter analysis strategy: dimensionality reduction reveals that text and visual tokens form distinct clusters, and analysis of two decoupled LoRA branches indicates strong orthogonality between their column spaces. The lower part illustrates our method: tokens are processed by separate branches where the Householder matrix $H^*$ acts on frozen orthogonal $\hat{Q}$, followed by learnable $\hat{R}$ modules to form the matrices $A$. The matrices $B$ are fully trainable, similar to the original LoRA.

by restricting updates to the dominant direction in column space. Koyun & Töreyin (2024) jointly exploits the row and column spaces for parameter sharing. Our method emphasizes not only efficient subspace utilization but also flexibility, enabling both high parameter efficiency and adaptability.

## 3 METHOD

In this section, we begin by analyzing the visual and text tokens in MLLMs, gradually presenting our research findings and the corresponding module designs to demonstrate the rationale behind our approach. The entire workflow of our method is illustrated in Figure 1.

### 3.1 PROBLEM ANALYSIS

**Decoupling parameters associated with visual and text tokens.** Currently, most MLLMs align visual tokens with the text token space of LLM backbones (Bai et al., 2025; Liu et al., 2023; Zhou et al., 2024). However, as mentioned in the Introduction, experiments in Figure 2 show that significant differences between visual and text tokens persist in model input.

Theoretically, parameter updates are closely related to the tokens. Associating parameters with tokens from different distributions leads to distinct characteristics in parameter matrices. Therefore, we aim to investigate what happens when we decouple the learnable parameters associated with visual and text tokens in MLLMs. To explore this, we first introduce independent reparameterization branches for the two token types, resulting in four parameter matrices:

$$A_{image}, A_{text} \in \mathbb{R}^{m \times r}, B_{image}, B_{text} \in \mathbb{R}^{r \times n}, \tag{1}$$

in which $m$, $n$ and $r$ represent the input dimension, output dimension, and LoRA rank, respectively. Let the visual and text tokens at the i-th layer be denoted as $t^i_{image}, t^i_{text} \in \mathbb{R}^{1 \times m}$, and the original parameter matrix of the model be $W \in \mathbb{R}^{m \times n}$. If the matrices $A$ and $B$ are multiplied as a whole, the forward propagation process can be expressed as:

$$t^{i+1}_{image} = t^i_{image}(W + A_{image}B_{image}) = t^i_{image}(W + W_{image}) \tag{2}$$

$$t^{i+1}_{text} = t^i_{text}(W + A_{text}B_{text}) = t^i_{text}(W + W_{text}). \tag{3}$$

In this way, the correlation between the learnable parameters associated with the two token types is significantly reduced during updates, leading to the relative decoupling of the matrices $W_{image}$

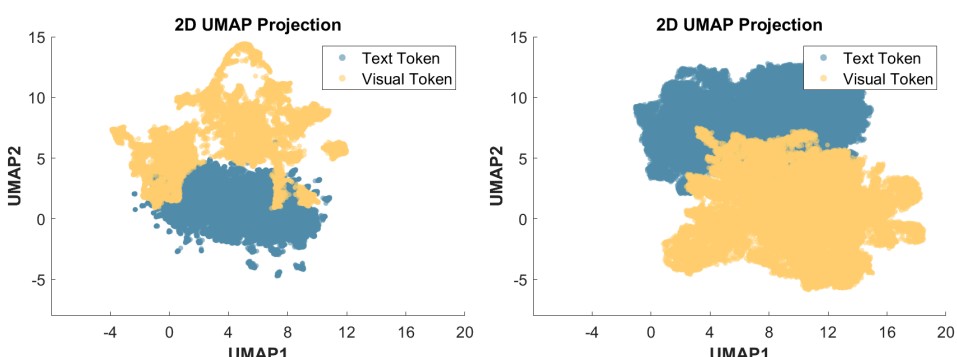

Figure 2: The distribution of visual and text embeddings in A-OKVQA (left) and TextVQA (right) datasets after 2D UMAP dimensionality reduction.

and $W_{text}$. After these decoupled parameter updates, the differences and relationships between the matrices $W_{image}$ and $W_{text}$ become the focus of our study.

**Exploring the subspace of the matrix as key feature in forward propagation.** Taking the original model's parameter matrix $W$ as an example, during forward propagation:

$$t^{i+1} = \{t^{i+1}_{[0]}, \ldots, t^{i+1}_{[n-1]}\} = t^i W = \{t^i W_{[0]}, \ldots, t^i W_{[n-1]}\}, \quad (4)$$

where $t^{i+1}_{[0]} \in \mathbb{R}$ is the element of $t^{i+1}$, and $W_{[0]} \in \mathbb{R}^n$ is the column vector of the matrix, essentially forming a basis for column space of the matrix. We can treat the multiplication between vector $t^i$ and matrix $W$ as a projection, where $t^i$ is projected onto each basis vector of the column space of $W$, resulting in each element of $t^{i+1}$. Ultimately, the resulting $t^{i+1}$ lies within the row space of matrix $W$. The projection directions, which correspond to the column vectors of $W$, indicate how the neural network determines the transformation of tokens within the feature space during forward propagation. Therefore, we believe that the column space of matrix $W$ is strongly correlated with the characteristics of input tokens $t$, so the differences between $t_{image}$ and $t_{text}$ may lead to differences between subspaces of $W_{image}$ and $W_{text}$.

**Using SVD to analyze matrix subspaces.** We use SVD to analyze properties of $W_{image}$ and $W_{text}$. Taking $W$ as an example, the SVD decomposition can be expressed as:

$$SVD(W) = U\Sigma V. \quad (5)$$

Here, matrix $\Sigma \in \mathbb{R}^{rank(W) \times rank(W)}$ contains the singular values, which are often the main focus in prior works (Zhang et al., 2023; Meng et al., 2024). In contrast, few studies consider the left/right singular vectors $U \in \mathbb{R}^{m \times rank(W)}$ and $V \in \mathbb{R}^{rank(W) \times n}$ matrices. In our case, analyzing $U$ and $V$ is more appropriate, as they form orthogonal bases that fully describe the row and column subspaces. Therefore, the relationships of $U_{image}\&U_{text}$ and $V_{image}\&V_{text}$ are central to our analysis. We also introduce a control group to represent the case where modality-specific parameters are not decoupled. It uses two separate reparameterization branches, but all tokens are passed through both. The corresponding parameter matrices are denoted as:

$$A_1, A_2 \in \mathbb{R}^{m \times r}, B_1, B_2 \in \mathbb{R}^{r \times n}. \quad (6)$$

The "angle" between two subspaces is a key indicator of their relationship, reflecting their relative positions. To compute this, we multiply the two sets of orthogonal bases and perform SVD. Take $V_{image}$ and $V_{text}$ as an example:

$$SVD(V^T_{image}V_{text}) = U_{it}\text{diag}(\sigma_{it[0]}, \sigma_{it[1]}, \ldots, \sigma_{it[r-1]})V_{it}. \quad (7)$$

The sum of squared singular values $\alpha = \|\{\sigma_{it[0]}, \sigma_{it[1]}, \ldots, \sigma_{it[r-1]}\}\|_2$ can serve as a measure of the cosine value of the angle between two subspaces, with larger values implying smaller angles and smaller values implying stronger orthogonality. (refer to the Appendix for more details). Using this, we compute the angles of $U_{image}\&U_{text}$, $V_{image}\&V_{text}$, $U_1\&U_2$, and $V_1\&V_2$. All of these

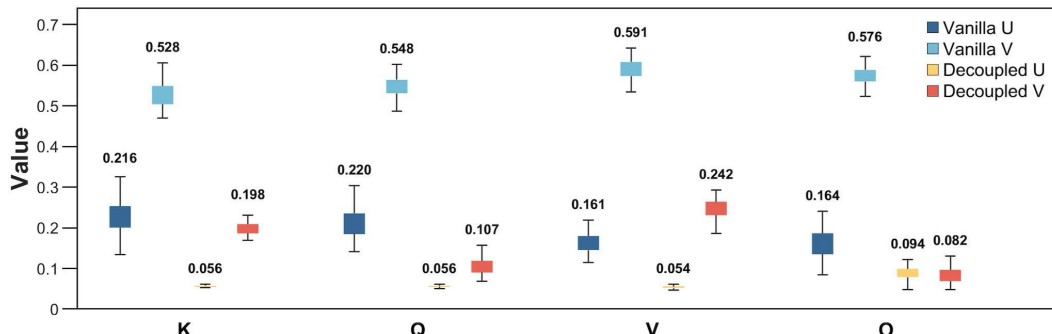

Figure 3: The cosine values of the angles between subspaces. The $U$ matrices are associated with the column spaces, while the $V$ matrices are associated with the row spaces. We compute four sets of results using the reparameterized matrices corresponding to the $K$, $Q$, $V$, and $O$ matrices in Attention blocks of each layer, and plot the results as boxplots.

matrices are obtained by performing SVD decomposition on the corresponding $W$ matrices. The computed cosine values of the angles between the subspaces are shown in Figure 3 as $Decoupled\,U$, $Decoupled\,V$, $Vanilla\,U$, and $Vanilla\,V$, respectively. We can clearly observe two findings. First, for both the Vanilla and Decoupled groups, the values corresponding to the $U$ matrices are consistently smaller than those of the $V$ matrices, indicating that the column spaces of the parameter matrices exhibit stronger orthogonality than the row spaces. Second, the values for the Decoupled group are significantly lower than those for the Vanilla group, demonstrating that modality decoupling yields much stronger subspace orthogonality, particularly in the column space. This outcome emerges after training, showing that under modality decoupling, tokens from different modalities tend to be projected onto highly orthogonal vectors during forward propagation. We believe this reflects how differences in modality-specific data characteristics shape network parameters, consistent with our earlier analysis.

## 3.2 MODEL ARCHITECTURE

**More parameter-efficient orthogonal basis vectors.** From the previous analysis, the column spaces of the $W_{image}$ and $W_{text}$ matrices exhibit high orthogonality, which stems from the orthogonality between the column spaces of the $A_{image}$ and $A_{text}$. Therefore, we can attempt to divide LoRA into separate branches for processing image and text tokens, while strengthening the orthogonality between the $A_{image}$ and $A_{text}$ matrices of the two branches. This approach aligns with the parameter characteristics and can reduce the covariance between modalities. The forward propagation equations remain as shown in equation 2 and equation 3. To maintain orthogonality between $A_{image}$ and $A_{text}$, we initialize both matrices orthogonally and freeze them to preserve orthogonality during model updates and reduce the number of trainable parameters to save memory.

**More flexible column vectors of the $A$ matrices.** While the above approach reduces the number of trainable parameters, it also decreases the flexibility of the module. The column vectors of the two $A$ matrices cannot be adaptively adjusted during training. Therefore, we further apply QR decomposition to $A$ matrices:

$$A_{image} = Q_{image}R_{image} = \{Q_{image}R_{image[0]}, Q_{image}R_{image[1]}, \ldots, Q_{image}R_{image[r-1]}\} \quad (8)$$

$$A_{text} = Q_{text}R_{text} = \{Q_{text}R_{text[0]}, Q_{text}R_{text[1]}, \ldots, Q_{text}R_{text[r-1]}\}. \quad (9)$$

The resulting $Q \in \mathbb{R}^{m \times r}$ matrices are orthogonal, describing a set of orthogonal bases for the original $A$ matrices respectively, and the $R \in \mathbb{R}^{r \times r}$ matrices represent the coordinates of the $A$ matrices' column vectors in this basis. To maintain orthogonality between the column spaces of $A_{image}$ and $A_{text}$, we simply need to ensure the perpendicularity of $Q_{image}$ and $Q_{text}$. Inspired by the form of QR decomposition, we replace matrices $A_{image}$ and $A_{text}$ with two frozen orthogonal basis matrices $\hat{Q}_{image}, \hat{Q}_{text} \in \mathbb{R}^{m \times \hat{r}}$ and learnable parameter matrices $\hat{R}_{image}, \hat{R}_{text} \in \mathbb{R}^{\hat{r} \times r}$. This allows the column vectors of the two $A$ matrices to vary freely within the space spanned by the orthogonal bases $Q$, improving flexibility. Since the learnable $\hat{R}$ matrices are small, they add little

overhead. To enhance expressiveness, we replace them with MLPs, still referring to them as $\hat{R}_{image}$ and $\hat{R}_{text}$ for simplicity.

**More flexible column space of the $A$ matrices.** While decomposing matrix $A$ increases the flexibility of its column vectors under limited trainable parameters, the frozen $Q$ matrix fixes the orthogonal basis at initialization. This indicates that the column vectors can vary within the space, but the column space itself remains fixed. To address this, we relax the QR constraint that the rank of $Q$ must match the rank $r$ of $A$, and set $\hat{r}$ as a tunable value. When $\hat{r} \gg r$, the column space of $\hat{Q}\hat{R}$ (i.e., of $A$), whose rank is smaller, can adaptively shift within the higher-rank space spanned by $\hat{Q}$, allowing the originally fixed column space to move freely within a larger range.

Meanwhile, even with a relatively large $\hat{r}$, the high-rank column space of $\hat{Q}$ remains a static subspace of the $m$-dimensional Euclidean space and is still limited in flexibility. To address this, we introduce Householder transformation to enable movement of the $\hat{Q}$ column space within the Euclidean space. Householder transformation is a mapping applied to vectors, defined as:

$$H = I - 2uu^T, \|u\|_2 = 1. \tag{10}$$

Multiplying any vector by the matrix $H$ results in its reflection across the hyperplane defined by $u$. Theoretically, the Householder transformation can model any rotation of the vector in space by changing $u$ and will preserve the relative position of a set of vectors when applied to all of them, which perfectly aligns with our requirements. Based on this, we design a learnable parameter vector $\hat{u}$, and apply the resulting $\hat{H}$ to multiply $\hat{Q}$. This enables column vectors of $\hat{Q}$ to move adaptively within the Euclidean space while maintaining relative orthogonality, with a minimal increase in parameters.

### 3.3 OVERVIEW OF THE DESIGN

The general architecture of our method is shown in Figure 1. In summary, our PEFT method can be expressed as follows:

$$t_{image}^{i+1} = t_{image}^i(W + H^*\hat{Q}_{image}\hat{R}_{image}B_{image}) \tag{11}$$

$$t_{text}^{i+1} = t_{text}^i(W + H^*\hat{Q}_{text}\hat{R}_{text}B_{text}). \tag{12}$$

Tokens of both modalities have their own independent parameter branches, with the original $A$ matrix decomposed into a frozen orthogonal matrix $\hat{Q}$ and a learnable module $\hat{R}$, and $H^*$ can be represented as:

$$H^* = \prod_{j=0}^{k-1} H^j = \prod_{j=0}^{k-1}(I - 2u^j u^{jT}). \tag{13}$$

This implies that multiple learnable Householder transformations will be applied to $\hat{Q}$ to enhance expressiveness. These designs achieve strict orthogonality between the column spaces of $A_{image}$ and $A_{text}$ with fewer learnable parameters, aligning with the characteristics of the inherent parameters and minimizing the covariance between the modalities. Meanwhile, the column spaces of the $A$ matrices retain the flexibility to fit any position in the Euclidean space, offering enhanced adaptability at minimal cost.

## 4 EXPERIMENTS

In this section, we describe our experimental setup, provide implementation details, and discuss the findings.

### 4.1 DATASETS AND BENCHMARKS

We compared our method with baselines on both general and domain-specific multimodal tasks. The general task datasets include A-OKVQA (Schwenk et al., 2022), OCR-VQA (Mishra et al., 2019), ScienceQA (Saikh et al., 2022), and TextVQA (Singh et al., 2019), covering common multimodal challenges such as OCR, image captioning, visual reasoning, and image understanding. Evaluations are conducted using the VLMEvalKit (Duan et al., 2024). For domain-specific tasks, we used

Table 1: The trainable parameters of each method in the experiments and their proportions relative to the full model parameters.

| Methods | SFT | AdaLoRA | LoRA | R-LoRA | VB-LoRA | VERA | Ours |
|---------|-----|---------|------|--------|---------|------|------|
| Params | 5.64 B | 56.63 M | 56.62 M | 54.39 M | 56.69 M | 1.05 M | 54.41 M |
| Percent | 74.45% | 0.74% | 0.74% | 0.71% | 0.74% | 0.01% | 0.70% |

Table 2: The results of general multimodal datasets. Methods marked with the * suffix are the best-performing methods, while those marked with the # suffix are the second-best-performing methods.

| Dataset | A-OKVQA | OCR-VQA | ScienceQA | TextVQA | - |
|---------|---------|---------|-----------|---------|---|
| Metrics | ACC | ACC±SD | ACC±SD | ACC | AVE-ACC |
| Pretrained Model | 83.06 | 61.04±6.84 | 72.98±27.25 | 65.07 | 70.94 |
| SFT | 83.93 | 62.86±7.65 | 87.36±15.71 | 68.72 | 75.72 |
| AdaLoRA | # 83.32 | 60.84± 7.85 | 78.09± 23.92 | 67.52 | 72.44 |
| LoRA | 83.14 | # 62.83±7.67 | # 88.30±15.57 | # 68.45 | # 75.68 |
| R-LoRA | 82.97 | 61.69±7.34 | 81.16±19.45 | 65.16 | 72.75 |
| VB-LoRA | 83.06 | 61.52±8.04 | 79.62±23.63 | 67.1 | 72.83 |
| VERA | 83.32 | 61.62±7.33 | 74.27±25.05 | 65.87 | 71.27 |
| Ours | * 84.02 | * 62.99±7.72 | * 90.48±15.13 | * 69.04 | * 76.63 |

VisOnlyQA (Kamoi et al., 2025), PathVQA (He et al., 2020), and Slake (Liu et al., 2021), which correspond to multimodal tasks for geometric problems in mathematics, biology, and medical imaging, respectively. Evaluation metrics include BLEU, ROUGE, and LLMSCORE. For efficiency, datasets exceeding 10k samples were truncated to 10k instances for training.

## 4.2 BASELINES AND IMPLEMENTATION DETAILS

We compared our method against the original pretrained model, full Supervised Fine-Tuning (SFT), and five PEFT baselines: AdaLoRA(Zhang et al., 2023), LoRA (Hu et al., 2022), VERA (Kopiczko et al., 2024), VB-LoRA (Li et al., 2024), and R-LoRA (Liu et al., 2025a). All experiments used LLaVA-1.6-Mixtral-7B as the backbone model, which is the latest version of LLaVA series (Liu et al., 2023). The random seed was fixed at $random\_seed = 42$, the LoRA rank was set at $r_{origin} = 32$, and reparameterization was applied to the following modules: $mlp.up\_proj$, $mlp.down\_proj$, and $mlp.gate\_proj$. The default learning rate was set to $5e-4$. For R-LoRA, which showed unstable convergence under this setting, an alternative learning rate $1e-4$ was used instead. Training was conducted on 4 NVIDIA A100 or A800 GPUs with Mixed Precision Training and DeepSpeed ZeRO-2 for acceleration. Due to designs of HydraLoRA and R-LoRA, we normalized their effective ranks to match the number of learnable parameters used by LoRA with $r_{origin} = 32$. Other PEFT baselines followed their official implementations. The trainable parameters and their proportion to the full model parameters are shown in Table 1.

For our method, two extra hyperparameters are introduced: $householder\_dim$ and $r_{scaling}$. The former controls the number of Householder transformations $k$, while the latter defines $\hat{r}$ as a scaled version of $r_{origin}$, i.e., $\hat{r} = r_{origin} \cdot r_{scaling}$. In our experiments, we set $householder\_dim = 3$ and $r_{scaling} = 6$, which ensured that the number of learnable parameters remained approximately aligned with LoRA.

## 4.3 EXPERIMENT RESULTS

### 4.3.1 GENERAL MULTIMODAL TASKS

In this set of experiments, we evaluated all methods on various general multimodal tasks, with results shown in Table 2. The ACC columns represent the correctness of the answers, evaluated by

Table 3: The results of domain-specific multimodal datasets. Methods marked with the * suffix are the best-performing methods, while those marked with the # suffix are the second-best performing methods.

| Datasets | PathVQA | | | Slake | | | VisOnlyQA | | |
|---|---|---|---|---|---|---|---|---|---|
| Metrics | BLEU | ROUGE | LS | BLEU | ROUGE | LS | BLEU | ROUGE | LS |
| Pretrained Model | 0.290 | 0.294 | 0.609 | 0.44 | 0.451 | 0.987 | 0.52 | 0.52 | 1.04 |
| SFT | 0.490 | 0.497 | 0.985 | 0.761 | 0.769 | 1.429 | 0.72 | 0.72 | 1.44 |
| AdaLoRA | 0.482 | 0.489 | 0.970 | 0.668 | 0.680 | 1.275 | # 0.72 | # 0.72 | # 1.44 |
| LoRA | # 0.496 | # 0.503 | # 0.994 | # 0.764 | # 0.773 | # 1.449 | 0.68 | 0.68 | 1.36 |
| R-LoRA | 0.332 | 0.337 | 0.703 | 0.759 | 0.767 | 1.441 | 0.71 | 0.71 | 1.42 |
| VB-LoRA | 0.480 | 0.485 | 0.960 | 0.691 | 0.703 | 1.315 | 0.55 | 0.55 | 1.10 |
| VERA | 0.353 | 0.358 | 0.725 | 0.496 | 0.504 | 1.002 | 0.57 | 0.57 | 1.14 |
| Ours | * 0.505 | * 0.512 | * 1.017 | * 0.767 | * 0.777 | * 1.452 | * 0.74 | * 0.74 | * 1.48 |

VLMEvalKit. And AVE-ACC column shows the average of ACC columns. The SD indicates the standard deviation of the OCR-VQA and ScienceQA subsets. As shown, nearly all methods outperform the original pretrained model across datasets, and our method achieves the most significant improvements. For example, on A-OKVQA, while other methods yield accuracy gains of only 0.1 to 0.3, ours improves by nearly 1.0. This demonstrates the strong performance of our approach on core multimodal tasks.

### 4.3.2 DOMAIN-SPECIFIC MULTIMAL TASKS

In this experiment, we constructed evaluation tasks using several domain-specific multimodal datasets to assess performance in specialized scenarios, and the results are shown in Table 3. We use BLEU and ROUGE to measure lexical similarity and additionally compute an LLMSCORE, where incorrect answers receive a score of 0, partially correct answers receive a score of 1, and fully correct answers receive a score of 2. The average score is then reported. Compared to general tasks, PEFT methods yield more noticeable improvements on domain-specific datasets. Our approach delivers the best and most stable performance across the domain-specific datasets, outperforming all baselines.

### 4.3.3 IMPACT OF RANK ON ACCURACY AND PARAMETER EFFICIENCY

To investigate the effect of matrix rank on performance, we fixed all other parameters and varied $r_{origin}$. The experiments were conducted on the A-OKVQA dataset, with the target modules set to $down\_proj$ for efficiency. We compared our method with LoRA as a baseline, testing $r_{origin}$ values from 10 to 60 with a step size of 10, as shown in Figure 4.

The left panel reports accuracy on the test set on different ranks. Both methods generally show improved performance as rank increases, consistent with the intuition that more learnable parameters yield higher expressiveness. Across most ranks, our method outperforms LoRA, demonstrating stronger stability. At higher ranks, our method achieves higher performance ceilings, reaching the best accuracy in this experiment at $r_{origin} = 60$. At very low ranks, e.g., $r_{origin} = 10$, LoRA performs more robustly, probably due to its simpler module structure. In our method, freezing parts of the matrix constrains the subspace, and with limited learnable parameters at low rank, expressiveness is restricted.

The right panel shows how the number of learnable parameters scales with rank. Under the current setting, our method consistently requires fewer learnable parameters than LoRA while achieving relatively better performance. Overall, our approach remains stable under rank variation and shows stronger potential at higher ranks.

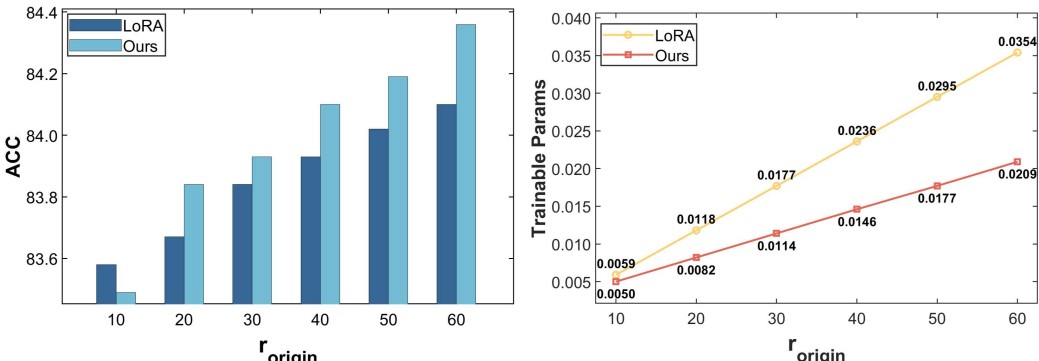

Figure 4: Comparison between our method and LoRA in terms of performance and the number of learnable parameters under different $r_{origin}$ values.

Table 4: The results of ablation study. Methods marked with the * suffix are the best-performing methods, while those marked with the # suffix are the second-best performing methods.

| Dataset | A-OKVQA | OCRVQA | ScienceQA | TextVQA | - |
|---|---|---|---|---|---|
| Metrics | ACC | ACC±SD | ACC ±SD | ACC | AVE-ACC |
| Decoupled | 82.36 | 62.01±7.98 | 88.15±16.73 | 68.74 | 75.32 |
| Frozen Orthogonal | 83.76 | 61.3±8.09 | 88.1±16.96 | 67.2 | 75.09 |
| QR KM-Init | 83.49 | 62.14±7.78 | 88.6±16.44 | 68.88 | 75.78 |
| QR Orth-Init | # 83.84 | # 62.83±7.86 | # 88.84±15.26 | # 68.89 | # 76.10 |
| Ours | * 84.02 | * 62.99±7.72 | * 90.48±15.13 | * 69.04 | * 76.63 |

## 4.4 ABLATION STUDY

The ablation results are shown in Table 4. The first row reports the performance of dual-branch LoRA with modality-decoupled parameters. The second row froze $A_{image}$ and $A_{text}$ after orthogonal initialization. In the third row, $A_{image}$ and $A_{text}$ were factorized by QR decomposition, with $\hat{Q}_{image}$ and $\hat{Q}_{text}$ initialized using Kaiming Uniform (He et al., 2015). The fourth row used the same structure as the third but adopted orthogonal initialization. The fifth row presents the complete version of our module, where a Householder transformation is further applied to the orthogonally initialized $\hat{Q}_{image}$ and $\hat{Q}_{text}$ matrices.

A comparison of the first and second rows shows that orthogonalizing and freezing the matrices leads to performance degradation. We attribute this to the reduction in learnable parameters, which limits the adaptability, and the cost of maintaining orthogonality outweighs its benefits. The fourth row introduces learnable parameters, making the column vectors more flexible, resulting in significant gains. However, when orthogonal initialization is abandoned (third row), performance decreases, indicating that the gains are due to both orthogonality and flexibility in the subspace. Finally, the Householder transformation further increases column space flexibility, yielding the highest performance. These experiments confirm that focusing on both subspace orthogonality and flexibility is essential and validate the effectiveness of each component in our design.

## 5 CONCLUSION

In this work, we propose a PEFT method tailored for MLLMs. Experiments demonstrate that our method achieves strong performance on both general and domain-specific multimodal tasks. Our findings highlight the importance of subspace orthogonality in multimodal PEFT, offering new perspectives on parameter design. Beyond performance gains, this may inspire more efficient and interpretable architectures for multimodal learning. More discussion about our method can be found in Appendix.

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

## A  APPENDIX

### A.1  LLM USAGE STATEMENT

We employed LLM to assist in our writing process, including grammar checking, wording refinement, and concise expression. We also used LLMs to support literature search and summarization, such as various PEFT methods and Related Works.

### A.2  THE CALCULATION OF THE ANGLE BETWEEN SUBSPACES

In the previous section, we calculated the angle between the subspaces of matrices. In this section, we will elaborate on the mathematical principles behind this calculation. We first present the overall computational process. Using the column spaces of $W_{image}$ and $W_{text}$ as an example, we first apply SVD to extract the orthogonal bases of both spaces:

$$SVD(W_{image}) = U_{image}\Sigma_{image}V_{image} \tag{14}$$
$$SVD(W_{text}) = U_{text}\Sigma_{text}V_{text}. \tag{15}$$

Here, $U_{image} \in \mathbb{R}^{m \times r}$ and $U_{text} \in \mathbb{R}^{m \times r}$ are orthogonal bases for the column spaces of $W_{image}$ and $W_{text}$, respectively. We then multiply these matrices and perform SVD again:

$$SVD(V_{image}^T V_{text}) = U_{it}\Sigma_{it}V_{it} = U_{it}\text{diag}(\sigma_{it[0]}, \sigma_{it[1]}, \ldots, \sigma_{it[r-1]})V_{it}. \tag{16}$$

The diagonal elements of the $\Sigma_{it}$ matrix represent the dot products between corresponding basis vectors of the two subspaces, reflecting the cosine of the angle between the corresponding dimensions. We use this to measure the angle between the two subspaces.

The principle is as follows: $U_{image}$ and $U_{text}$ are orthogonal bases for the column spaces of the two matrices. Since there are infinite vectors in a space, measuring the angle between arbitrary

vectors is not suitable. However, an orthogonal basis fully represents a vector space, so we believe using orthogonal bases to represent the relationship between two spaces is more appropriate. Let orthogonal matrices $K_{image} \in \mathbb{R}^{r \times r}$ and $K_{text} \in \mathbb{R}^{r \times r}$ be introduced, such that:

$$U'_{image} = U_{image} K_{image} \tag{17}$$

$$U'_{text} = U_{text} K_{text}. \tag{18}$$

Thus, $U'_{image}$ and $U'_{text}$ remain orthogonal bases for the column spaces of $W_{image}$ and $W_{text}$, respectively, and vary according to the values of $K_{image}$ and $K_{text}$. Therefore, in theory, they can represent any orthogonal basis for the spaces. Naturally, we compute the dot product between the vectors of the two orthogonal bases by multiplying the matrices $U'_{image}$ and $U'_{text}$:

$$U_{mult} = U'^{T}_{image} U'_{text} = K^{T}_{image} U^{T}_{image} U_{text} K_{text}. \tag{19}$$

Here, the element at position $(i, j)$ in $U_{mult}$ corresponds to the dot product between the $i$-th vector in $U'_{image}$ and the $j$-th vector in $U'_{text}$. $U_{mult}$ represents the result of taking the dot product of each pair of vectors from the two sets of orthogonal bases, which partially reflects the angle between the two subspaces. However, $U_{mult}$ will change with different values of $K_{image}$ and $K_{text}$, leading to inconsistencies in the measure.

Therefore, we need to find two sets of orthogonal bases that most clearly represent the positional relationship between the two subspaces. We perform SVD on $U^{T}_{image} U_{text}$:

$$SVD(U^{T}_{image} U_{text}) = U_{it} \Sigma_{it} V_{it}. \tag{20}$$

Here, $U_{it} \in \mathbb{R}^{r \times r}$ and $V_{it} \in \mathbb{R}^{r \times r}$ are orthogonal matrices, and $\Sigma_{it} \in \mathbb{R}^{r \times r}$ is a diagonal matrix. Now, let:

$$K^{T}_{image} = U^{-1}_{it} \tag{21}$$

$$K_{text} = V^{-1}_{it}. \tag{22}$$

This leads to:

$$U_{mult} = U'^{T}_{image} U'_{text} = K^{T}_{image} U^{T}_{image} U_{text} K_{text} = \Sigma_{it}. \tag{23}$$

Thus, we have found two sets of orthogonal bases, $U'_{image}$ and $U'_{text}$, whose matrix product results in a diagonal matrix $\Sigma_{it}$. This means that for any vector $U'_{image[i]}$ in $U'_{image}$, the dot product with the corresponding vector $U'_{text[i]}$ in $U'_{text}$ is $\Sigma_{it[i]}$, and the dot product with all other vectors is zero. Therefore, we consider that the vectors in the two sets of orthogonal bases, $U'_{image}$ and $U'_{text}$, are "aligned" in some sense, meaning that they have angles only with their corresponding vectors and are orthogonal to all other vectors. We believe that the diagonal elements of $\Sigma_{it}$ clearly represent the positional relationship between the dimensions of the two spaces, and the sum of their squared values provides a measure of the overall angle between the two subspaces.

### A.3 SUPPLEMENTARY EXPERIMENTS

#### A.3.1 THE EFFECT OF ANGLE BETWEEN $\hat{A}_{image}$ AND $\hat{A}_{text}$

We set up a control experiment to explore the impact of the angle between the column spaces of two $\hat{A}$ matrices on algorithm performance. We first initialize the matrix $\hat{A}_{text}$ and generate an orthogonal matrix $\hat{A}^{\perp}_{text}$ to it. We perform interpolation using the following equation and apply QR decomposition:

$$A_{inter} = \gamma \hat{A}^{\perp}_{text} + (1 - \gamma) \hat{A}_{text} = Q_{inter} R_{inter} \tag{24}$$

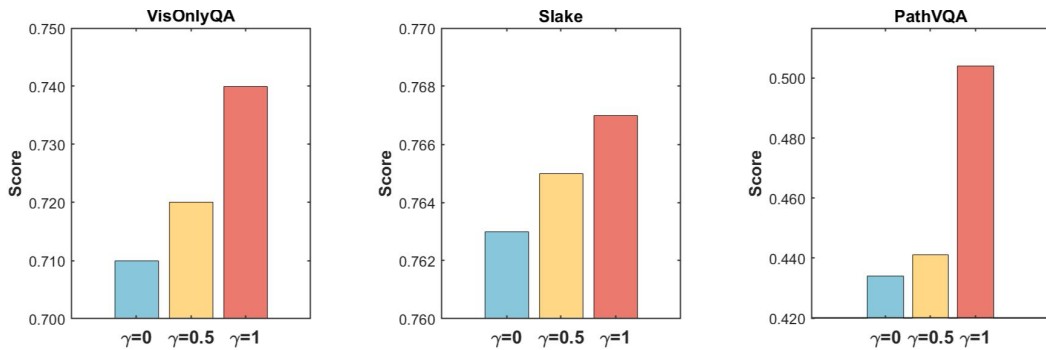

Figure 5: The results of different column space angles.

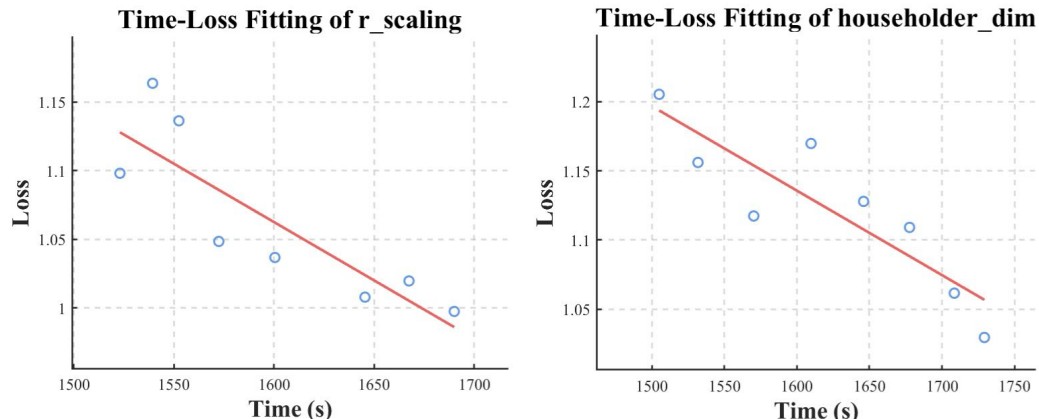

Figure 6: Time-Loss Fitting of hyperparameters $r\_scaling$ and $householder\_dim$.

where $Q_{inter}$ and $R_{inter}$ are the QR decomposition results, and $\gamma$ is the interpolation coefficient. We set $\gamma$ to 0, 0.5, and 1, and initialize $\hat{A}_{image}$ with the resulting orthogonal matrix $Q_{inter}$. When $\gamma = 0$ or 1, $Q_{inter}$ corresponds to $\hat{A}_{text}$ or $\hat{A}_{text}^{\perp}$, respectively. Since they are orthogonal matrices, the $Q$ matrix of the QR decomposition is simply the matrix itself. When $\gamma = 0.5$, the column space of $Q_{inter}$ is between complete overlap with $\hat{A}_{text}$ and complete orthogonality.

In this way, we obtain three sets of matrices with decreasing orthogonality between their column spaces. We then tested the performance of our method under these three initializations, which is illustrated in Figure 5. It is evident that as the orthogonality between the column spaces increases, the algorithm performs better. We believe that this reflects the positive impact brought by the reduction of covariance, which is caused by orthogonality between the two modality tokens.

### A.3.2    HYPERPARAMETER ANALYSIS

In this section, we tested two hyperparameters of our method, $r\_scaling$ and $householder\_dim$, on the A-OKVQA dataset. $r\_scaling$ scales the rank $r_{origin}$ of the matrix $\hat{Q}$:

$$\hat{r} = r\_scaling \times r_{origin}. \tag{25}$$

Here, $\hat{r}$ represents the dimension of the orthogonal space spanned by the $\hat{Q}$ matrix. $householder\_dim$ represents the number of Householder transformations applied.

We conducted two sets of experiments. In the first, we fixed $householder\_dim$ and varied $r\_scaling$ from 1 to 29 in steps of 4, yielding eight results. In the second, we fixed $r\_scaling$ and varied $householder\_dim$ from 1 to 15 in steps of 2. We plotted the training loss against training time for each result, using a line fit to explore how $r\_scaling$ affects the fitting of the model and the training

speed. The results, shown in Figure 6, reveal that as $r\_scaling$ or $householder\_dim$ increases, the training speed increases steadily. This is due to the increased rank of the $\hat{Q}$ matrix as $r\_scaling$ rises, adding more parameters, while larger $householder\_dim$ will introduce higher computational cost due to more times of Householder transformations. Training loss decreases with both parameters, but not strictly monotonically, which we believe may be related to the choice of dataset and random seed.

### A.4 LIMITATION AND FUTURE WORK

Due to current hardware constraints, our experiments are limited to medium-scale models, and further validation is needed on more domain-specific tasks as well as additional modalities such as speech and video. Moreover, although we observed that modality orthogonality benefits the tasks studied in our experiments, it remains unclear whether strict orthogonality may impose excessive constraints and thereby limit expressiveness in broader tasks, which requires further investigation. In future work, we plan to explore the impact of orthogonality on a wider variety of tasks and extend our framework to additional modalities. We also aim to integrate our design with other types of PEFT methods to further assess its feasibility and generalizability.

