# OpenReview forum: "ORFLEX: Orthogonal Reparameterization with Flexibility for Multimodal Large Language Model Fine-Tuning"
_ICLR.cc/2026/Conference — ICLR 2026 Conference Withdrawn Submission_

### Official Review · Reviewer_mUpE · 2025-10-25

**Soundness:** 2
**Presentation:** 2
**Contribution:** 3
**Rating:** 4
**Confidence:** 3

**Summary:**

This paper proposes ORFLEX, aorthogonal reparameterization method for PEFT  of MLLMs.
The authors observe that LoRA subspaces corresponding to visual and textual modalities are strongly orthogonal when decoupled.
Building on this insight, they introduce modality-specific reparameterization branches with a QR-inspired decomposition, splitting LoRA matrices into a frozen orthogonal basis and a lightweight learnable matrix.
They further enhance flexibility by incorporating learnable Householder transformations* which adaptively rotate the orthogonal subspace without violating orthogonality.
Experiments on both general and domain-specific multimodal benchmarks (e.g., A-OKVQA, ScienceQA, PathVQA) show consistent improvements over strong PEFT baselines such as LoRA, AdaLoRA, and VB-LoRA, while maintaining comparable or lower trainable parameter counts.

**Strengths:**

1. The work introduces an orthogonality-based view of modality decoupling in PEFT, combining QR decomposition and Householder transformations in a novel way.

2. Theoretical justification for orthogonal subspace alignment is solid, and the empirical validation across datasets is comprehensive.

3. The authors did comprehensive analysis, including ablation, rank-scaling, and hyperparameter studies that clearly support design choices.

**Weaknesses:**

1. Experiments are restricted to mid-sized MLLMs (LLaVA-1.6-Mixtral-7B); results on larger models (e.g., Qwen3-VL) would strengthen generalizability. It is hard to know the generalization ability of the method across different backbone models.

2. The paper notes that enforcing strong orthogonality might limit expressiveness for tasks requiring deep modality fusion; this tradeoff is not fully explored.

3. Although authors claims they use fewer trainable parameter,  but training time and GPU memory usage comparisons to LoRA and AdaLoRA aren't reported. It would help quantify the claimed efficiency benefits.

**Questions:**

1. How sensitive is ORFLEX to the number of Householder transformations? Could too many lead to instability or overfitting?

2. Can ORFLEX generalize to non-vision modalities (e.g., audio-text or video-text), or is its design specific to vision-language alignment?

3. Could the authors analyze computational overhead more directly (e.g., FLOPs or throughput) relative to LoRA?

---

> ### Author Response · Authors · 2025-11-25
>
> ## 1. Testing with Larger Models (Weakness1)
>
> We attempted to validate our method on a larger model, InternVL3.5-14B, and obtained the following results. The results show that our method performs well on larger models.
>
> |         | A-OKVQA | OCR-VQA | TextVQA | AVERAGE-ACC |
> | :-----: | :-----: | :-----: | :-----: | :---------: |
> | AdaLoRA |  86.91  |  82.42  |  68.85  |    79.39    |
> |  LoRA   |  87.70  |  84.96  |  69.33  |    80.66    |
> | VB-LoRA |  86.13  |  84.86  |  68.95  |    79.98    |
> |  VERA   |  87.40  |  82.03  |  42.48  |    70.64    |
> |  PiSSA  |  87.79  |  84.96  |  68.94  |    80.56    |
> |  Ours   |  88.18  |  85.64  |  69.63  |    81.09    |
>
> ## 2. Discussion on the Trade-off Between Orthogonality and Expressiveness (Weakness2)
>
> The trade-off between orthogonality and expressiveness has not yet been systematically characterized in the literature for multimodal tasks requiring deep modality interactions. This concern mainly arises from discussions in theoretical research on simple neural networks, where orthogonality's advantages include increased stability in forward propagation and gradient training, as well as improved generalization [1, 2]. At the same time, some studies point out that in cases of strong orthogonality, it may limit the representational ability of simpler neural networks with fewer layers [3]. From a function class perspective, suppose the function class with orthogonality constraints is:
> $$
> F_{orth} = \\{ f_{\theta} : \theta = (W^{(1)}, \ldots, W^{(L)}), W^{(l)\top} W^{(l)} = I \\}.
> $$
> Clearly, we have
> $$
> F_{\text{orth}} \subset F_{\text{unconstrained}}.
> $$
> Orthogonality constraints lead to more stable and less redundant representations, but they also shrink the representable function class, thus potentially creating a trade-off between expressiveness and the strength of regularization.
>
> In multimodal scenarios, we believe that the gain from orthogonality lies in improving the problem of modality information loss. It helps preserve modality-specific information and reduces redundancy between features. However, similar to the effect of orthogonal operations on simple neural networks, there might also be concerns about a reduction in the expressiveness of the representable function class, which could affect modality fusion. Nevertheless, we remain optimistic about this issue, mainly because:
>
> 1. For large models, the structure of the model is more complex, and its expressiveness and flexibility are much stronger than simple neural networks. Many studies related to large models have incorporated direct orthogonalization in their algorithm design, and results show that it does not significantly negatively impact model performance [4, 5, 6].
>
> 2. A key innovation of our network design is that it maintains orthogonality while allowing flexibility in the parameter space to avoid the limitations caused by simple orthogonalization. The comparison between the second row (Frozen Orthogonal) and the last row in Table 4 of the main text illustrates the effectiveness of our design.
>
> 3. Existing studies show that multimodal large model fusion mainly focuses on the Attention Module [7], while PEFT methods primarily affect other modules like MLP, and have limited impact on modality fusion. Additionally, it is important to clarify that in our method, orthogonality is applied only to the reparameterized matrices, while the original matrices of the model remain shared between modalities. For more details, please refer to our discussion with Reviewer ALFh in Topic 4. Therefore, there is no significant disruption to modality fusion due to absolute orthogonality.
>
> [1] Vorontsov, Eugene, et al. "On orthogonality and learning recurrent networks with long term dependencies." *International Conference on Machine Learning*. PMLR, 2017.
>
> [2] Bansal, Nitin, Xiaohan Chen, and Zhangyang Wang. "Can we gain more from orthogonality regularizations in training deep networks?." *Advances in Neural Information Processing Systems* 31 (2018).
>
> [3] Huang, Lei, et al. "Orthogonal weight normalization: Solution to optimization over multiple dependent stiefel manifolds in deep neural networks." *Proceedings of the AAAI Conference on Artificial Intelligence*. Vol. 32. No. 1. 2018.
>
> [4] Zhang, Xin, et al. "An Orthogonal High-Rank Adaptation for Large Language Models." *Proceedings of the 2025 Conference on Empirical Methods in Natural Language Processing*. 2025.
>
> [5] Qiu, Zeju, et al. "Reparameterized LLM Training via Orthogonal Equivalence Transformation." *arXiv preprint arXiv:2506.08001* (2025).
>
> [6] Gorbunov, Mikhail, et al. "Group and shuffle: Efficient structured orthogonal parametrization." *Advances in neural information processing systems* 37 (2024): 68713-68739.
>
> [7] Ma, Feipeng, et al. "Ee-mllm: A data-efficient and compute-efficient multimodal large language model." *arXiv preprint arXiv:2408.11795* (2024).

---

> ### Author Response · Authors · 2025-11-25
>
> ## 3. Comparison of Training Costs with Baselines (Weakness3, Problem3)
>
> The training costs for each baseline are shown in the table below. It can be seen that the additional complexity introduced by our method is acceptable. The memory and time costs for the baselines are higher than ours. Although our method appears to have several modules, the number of trainable parameters has not increased. We only introduced some frozen parameters and Householder transformations, which increased the FLOPs. The impact on time/memory cost is acceptable.
>
> |            | AdaLoRA | LoRA  | R-LoRA | VB-LoRA | VERA  | DoRA  | PiSSA | Ours  |
> | :--------: | :-----: | :---: | :----: | :-----: | :---: | :---: | :---: | :---: |
> | GPU Memory |  40.81  | 40.73 | 43.60  |  40.82  | 41.16 | 43.63 | 40.74 | 43.57 |
> | time/step  |  5.85   | 5.75  |  5.75  |  6.06   | 5.71  | 5.24  | 5.49  | 5.99  |
> |   flops    |  59.47  | 59.61 | 59.68  |  59.61  | 59.61 | 59.79 | 59.61 | 59.99 |
>
> ## 4. Sensitivity of the Number of Householder Transformations (Problem1)
>
> In Supplement Section A.3.2, we performed scaling experiments on the number of Householder transformations, setting the maximum number of transformations to 15. This is a high value because, at this point, the learnable parameters involved in Householder transformations are comparable to those of a LoRA matrix with rank = 8. In this case, we observed that the loss still remains within a reasonable range and does not show extreme behavior.
>
> Therefore, we believe that a large number of Householder transformations does not lead to instability. The learnable parameters outside of the Householder transformations already provide relatively good robustness, and the overall stability of the system is not significantly affected by the number of Householder transformations, whether it is too high or too low.
>
> ## 5. Generalization to Non-Image Modalities (Problem2)
>
> In multimodal problems, different modalities generally contain unique information that is different from other modalities. This is a common feature, so we believe that our method can be generalized to any modality.
>
> We first conducted additional experiments on the audio modality. For the video modality, due to the higher resource requirements, experiments are still ongoing, and we will report the results once they are available. We believe the results will be similar to those of the image modality. For the audio modality, we used Qwen2-Audio-7B-Instruct as the base model and obtained the following results. As can be seen, our method also performs well on the audio modality.
>
> |            | AdaLoRA | LoRA  | R-LoRA | VB-LoRA | VERA  | DoRA  | PiSSA | Ours  |
> | ---------- | ------- | ----- | ------ | ------- | ----- | ----- | ----- | ----- |
> | Clotho-AQA | 52.73   | 58.20 | 57.32  | 49.90   | 50.59 | 58.11 | 57.62 | 59.47 |
>
> Additionally, we performed orthogonality analysis on more datasets and multiple modalities (text-image, text-audio) as done in the main text. The results are as follows. As can be seen, orthogonality between parameter matrix subspaces remains evident across modalities and datasets.
>
> |                 | A-OKVQA | OCR-VQA | ScienceQA | TextVQA | Clotho-AQA |
> | :-------------: | :-----: | :-----: | :-------: | :-----: | :--------: |
> |  $Q_{vanilla}$  |  0.220  |  0.193  |   0.158   |  0.136  |   0.176    |
> | $Q_{decoupled}$ |  0.056  |  0.054  |   0.057   |  0.055  |   0.058    |
> |  $K_{vanilla}$  |  0.216  |  0.189  |   0.162   |  0.137  |   0.178    |
> | $K_{decoupled}$ |  0.056  |  0.054  |   0.056   |  0.056  |   0.059    |
> |  $V_{vanilla}$  |  0.161  |  0.165  |   0.143   |  0.137  |   0.153    |
> | $V_{decoupled}$ |  0.054  |  0.055  |   0.056   |  0.055  |   0.057    |
> |  $O_{vanilla}$  |  0.164  |  0.162  |   0.145   |  0.135  |   0.149    |
> | $O_{decoupled}$ |  0.093  |  0.078  |   0.081   |  0.084  |   0.080    |

---

### Official Review · Reviewer_ALFh · 2025-10-26

**Soundness:** 2
**Presentation:** 2
**Contribution:** 2
**Rating:** 4
**Confidence:** 4

**Summary:**

The paper proposes a modality-decoupled orthogonal reparameterization of LoRA that preserves inter-modality column-space orthogonality via frozen orthonormal bases, learnable coefficients, and Householder rotations, enabling robust MLLM fine-tuning at near-LoRA cost.

**Strengths:**

The paper is clearly written and well-organized, with fluent language that make the presentation easy to follow.

**Weaknesses:**

1. The logic of the paper’s motivation is unclear. It states that “despite aligning visual tokens with language token feature spaces, discrepancies between modalities persist during the forward pass, which current LoRA-based methods overlook.” However, this issue also seems to exist under full fine-tuning for MLLMs and therefore cannot be said to arise specifically from using LoRA.

2. The paper introduces a series of components that appear to have high time complexity. Please compare training wall-clock time and GPU memory usage against baselines such as LoRA.

3. In Tables 2 and 3, the improvements over LoRA are modest. Please explain why other baselines underperform LoRA. Could the authors also compare with stronger baselines relative to LoRA, such as DoRA [1] and PiSSA [2]?

4. It is unclear whether the orthogonality operation in the method makes sense; judging from the ablation study, the gains also appear limited. Could the authors provide further insights into enforcing orthogonality for LoRA across different modalities?

[1] DoRA: Weight-Decomposed Low-Rank Adaptation

[2] PiSSA: Principal Singular Values and Singular Vectors Adaptation of Large Language Models

**Questions:**

The SD in Table 2 seems large. Is this because the dataset subsets differ substantially?

Since the proposed method’s gains appear limited, there is no significant parameter efficiency, and it can only be applied to multimodal fine-tuning, comparisons with stronger baselines (DoRA, PiSSA, etc.) are necessary.

---

> ### Author Response · Authors · 2025-11-25
>
> Thank you for your valuable suggestions and your patience! We will address your concerns point by point. If you have any unresolved questions, please feel free to continue asking!
>
> ## 1. Explanation of the Motivation Logic in the Paper (Weakness1)
>
> For the motivation statement in our paper, we have two main explanations:
>
> 1. **The low-rank nature of LoRA makes it much harder to handle modality differences compared to full SFT.**
>
> For full SFT, since it completely inherits the pre-training approach and has strong generalization, it can easily handle modality differences in pre-trained models. On the other hand, LoRA, due to its low-rank matrix characteristics, may have limited ability to handle modality differences. We can analyze this from a matrix theory perspective. Suppose during forward propagation, the visual and text embeddings are represented as $x_v$ and $x_t$, and the ideal transformed values are $x_v^{'}$ and $x_t^{'}$. For full SFT, the conditions it needs to satisfy are:
> $$
> x_v^{'}=Ax_v
> $$
> $$
> x_t^{'}=Ax_t
> $$
> This can be viewed as two systems of equations:
> $$
> x_v^{'}=Ax
> $$
> $$
> x_t^{'}=Ay
> $$
> We need to find an $A$ such that both systems have solutions, with $x_v$ being in the solution space of the first equation and $x_t$ in the solution space of the second equation.
>
> Let the null space of matrix $A$ be $\mathbb{A}^n$, with a rank of $rank(A)$, and let the particular solutions of the above two systems be $\xi_1$ and $\xi_2$. Then the solution spaces of both are translations of $\mathbb{A}^n$ towards $\xi_1$ and $\xi_2$. In full SFT, $A$ is a full-rank matrix, so the dimension of the solution space is large, making it easier to satisfy the condition that $x_v$ and $x_t$ are in the solution space. In contrast, for LoRA, the rank of $A$ is usually one or two orders of magnitude smaller than full-rank, significantly reducing the size of the solution space and making it harder to satisfy the condition.
>
> In our method, we decouple the related parameters of visual and text embeddings:
> $$
> x_v^{'}=Bx_v
> $$
> $$
> x_t^{'}=Cx_t
> $$
> Let the null spaces of matrices $B$ and $C$ be $\mathbb{B}^n$ and $\mathbb{C}^n$, respectively. In this case, the solution space of the system is no longer a translation from $\mathbb{A}^n$, but rather from two independent subspaces, $\mathbb{B}^n$ and $\mathbb{C}^n$. This clearly makes it easier to satisfy the conditions in the low-rank case.
>
> 2. **PEFT methods offer high design flexibility, but LoRA has not fully exploited its potential.**
>
> Modality differences always exist in pre-training and SFT for MLLM, and addressing this feature in pre-trained models is a costly design task. However, PEFT methods offer a high degree of design freedom at a low cost, making them ideal for leveraging modality differences. Unfortunately, existing PEFT methods have not fully exploited this feature. Therefore, our motivation includes the intention to fill this gap by designing a better structure that better captures modality differences.
>
> ## 2. Comparison of Training Costs with Baselines (Weakness2)
>
> The training costs for each baseline are shown in the table below. It can be seen that the additional complexity introduced by our method is acceptable. The memory and time costs for the baselines are higher than ours. Although our method appears to have several modules, the number of trainable parameters has not increased. We only introduced some frozen parameters and Householder transformations, which increased the FLOPs. The impact on time/memory cost is acceptable.
>
> |            | AdaLoRA | LoRA  | R-LoRA | VB-LoRA | VERA  | DoRA  | PiSSA | Ours  |
> | :--------: | :-----: | :---: | :----: | :-----: | :---: | :---: | :---: | :---: |
> | GPU Memory |  40.81  | 40.73 | 43.60  |  40.82  | 41.16 | 43.63 | 40.74 | 43.57 |
> | time/step  |  5.85   | 5.75  |  5.75  |  6.06   | 5.71  | 5.24  | 5.49  | 5.99  |
> |   flops    |  59.47  | 59.61 | 59.68  |  59.61  | 59.61 | 59.79 | 59.61 | 59.99 |
>
> ## 3. Adding DoRA and PiSSA as Baselines (Weakness3)
>
> We added DoRA and PiSSA as additional baselines. We unified key parameters such as rank and target_modules for all methods, and the performance on various datasets is shown in the table below. It can be seen that the performance of these two methods is still comparable to LoRA. We believe this may be related to the differences between multimodal tasks and text-only tasks, as many baselines have not been validated on multimodal tasks. Additionally, some baselines, such as R-LoRA, are based on empirical analysis that does not account for multimodal problems.
>
> |       | A-OKVQA | OCR-VQA | ScienceQA | TextVQA | AVERAGE-ACC |
> | :---: | :-----: | :-----: | :-------: | :-----: | :---------: |
> | LoRA  |  83.14  |  62.83  |   88.30   |  68.45  |    75.68    |
> | PiSSA |  83.58  |  62.70  |   88.00   |  68.46  |    75.69    |
> | DoRA  |  83.41  |  62.60  |   87.06   |  68.57  |    75.41    |
> | Ours  |  84.02  |  62.99  |   90.48   |  69.04  |    76.63    |

---

> ### Author Response · Authors · 2025-11-25
>
> ## 4. Further Explanation of the Rationale Behind Orthogonality Operation (Weakness4)
>
> Orthogonality in multimodal problems helps eliminate the extra redundancy between the different feature dimensions of embeddings. We believe this is beneficial for capturing modality-specific information and resolving the issue of information confusion when modalities are processed indiscriminately. Some existing works have also mentioned this point [1, 2, 3]. In our method, orthogonality only acts on the reparameterized matrices, while the original matrices of the model are modality-shared. This gives the model an advantage when processing information patterns that are similar across two modalities. This results in the model's original matrix encoding fundamental information, while the orthogonal reparameterized matrix focuses more on encoding modality-independent information, supplementing the original matrix with a more optimal encoding scheme. The correlation data between two modalities that we presented in our discussion with Reviewer rEFZ in Topic 3 provides a good proof of this.
>
> Specifically, consider the forward propagation process of the LoRA model:
> $$
> t^{i+1} = t^i (W + A B) = t^i (W + W_{AB})
> $$
> where $t$ represents the embedding vector, and $W$ and $AB$ represent the original model matrix and LoRA matrix, respectively. $W_{AB}$ must simultaneously adapt to the information patterns in both text and visual embeddings, which we believe is a challenge for a low-rank matrix.
>
> In our method, forward propagation is split into two branches:
> $$
> t_{image}^{i+1} = t_{image}^i (W + A_{image} B_{image}) = t_{image}^{i}W + t_{image}^{i}W_{image}
> $$
>
> $$
> t_{text}^{i+1} = t_{text}^i (W + A_{text} B_{text}) = t_{text}^i W + t_{text}^iW_{text}
> $$
>
> The orthogonality in our method exists between the column spaces of the reparameterized matrices $A_{image}$ and $A_{text}$, which better removes redundancy between different feature dimensions of the embeddings and more effectively adapts and extracts modality-specific information to be incorporated into $t^{i+1}$.
>
> Additionally, as mentioned in the main text, we found that modules with high orthogonality can actively maintain orthogonality, which can save a significant amount of learnable parameters. We believe this is a better use of resources.
>
> [1] Zhao, Dongfang. "Approximate Fiber Product: A Preliminary Algebraic-Geometric Perspective on Multimodal Embedding Alignment." *arXiv preprint arXiv:2412.00373* (2024).
>
> [2] Braman, Nathaniel, et al. "Deep orthogonal fusion: multimodal prognostic biomarker discovery integrating radiology, pathology, genomic, and clinical data." *International conference on medical image computing and computer-assisted intervention*. Cham: Springer International Publishing, 2021.
>
> [3] Sun, Haoqin, et al. "Fine-grained disentangled representation learning for multimodal emotion recognition." *ICASSP 2024-2024 IEEE International Conference on Acoustics, Speech and Signal Processing (ICASSP)*. IEEE, 2024.
>
> ## 5. Is the Large Standard Deviation in Table 2 Due to Significant Differences Between Subsets? (Problem1)
>
> Yes, the subsets of the dataset belong to different domains of problem images, and some domains may be more difficult for the model, while others are relatively simple, leading to a larger standard deviation.

---

### Official Review · Reviewer_rEFZ · 2025-10-28

**Soundness:** 2
**Presentation:** 2
**Contribution:** 3
**Rating:** 4
**Confidence:** 3

**Summary:**

This work proposes a new PEFT framework designed specifically for MLLMs. The authors identify that existing LoRA-based PEFT approaches fail to handle the modality discrepancies between visual and textual tokens, whose embeddings and attention patterns remain partially disjoint during forward propagation. To address this, ORFLEX introduces modality-specific reparameterization branches and decomposes each LoRA matrix via a QR-inspired factorization into a frozen orthogonal basis and a lightweight learnable matrix. To further enhance adaptability, learnable Householder transformations are incorporated to rotate the orthogonal basis​ while preserving orthogonality, balancing efficiency and flexibility. Empirically, ORFLEX consistently outperforms state-of-the-art PEFT baselines such as LoRA, AdaLoRA, and VB-LoRA on multiple multimodal benchmarks (A-OKVQA, TextVQA, ScienceQA, PathVQA, Slake, etc.) while using only about 0.7% of the trainable parameters of full fine-tuning.

**Strengths:**

The methodology is well grounded in matrix theory, supported by rigorous analysis of subspace orthogonality using SVD, and validated by comprehensive ablation studies. The paper demonstrates both theoretical soundness and empirical rigor, with consistent performance improvements across diverse benchmarks. The orthogonal–flexible balance is thoroughly examined through quantitative experiments and controlled analyses, confirming the claimed benefits.

Extensive experiments on both general-purpose (A-OKVQA, TextVQA, ScienceQA) and domain-specific (PathVQA, Slake, VisOnlyQA) benchmarks provide strong evidence of the method’s robustness and scalability. The parameter efficiency (only 0.7% trainable parameters) with superior or comparable accuracy to full fine-tuning demonstrates high practical value.

**Weaknesses:**

1. Unclear interpretation of orthogonal embeddings: The paper claims that visual and textual embeddings exhibit strong orthogonality, yet this contradicts the common assumption that well-trained multimodal representations should be semantically aligned. If the embeddings are nearly orthogonal, the cosine similarity between corresponding visual and textual features would approach zero, undermining cross-modal semantic consistency. The authors should clarify this apparent paradox—whether “orthogonality” refers to parameter subspace independence rather than embedding-level semantic dissimilarity—and provide empirical evidence or visualization to support this interpretation.

2. Limited theoretical depth in orthogonality justification: While the paper empirically verifies that modality-specific LoRA subspaces exhibit orthogonality, the theoretical reasoning for why such orthogonality leads to improved generalization or reduced covariance is relatively shallow. A more formal derivation or theoretical framework (e.g., based on linear subspace interaction or feature disentanglement) would strengthen the conceptual foundation.

2. Lack of downstream interpretability or visualization: Although the paper discusses orthogonality in subspaces, it does not provide visual or interpretive evidence (e.g., token projection visualizations, correlation matrices) showing how the reparameterized subspaces improve modality separation or feature alignment. Adding such qualitative analysis would enhance interpretability and strengthen the intuitive understanding of the method’s effect.

3. Additional ablation needed for fairness verification: The paper should include an experiment where image and text branches use different low-rank matrices trained independently (without the orthogonal constraint) to directly compare against ORFLEX. This would reveal whether the observed gains truly come from orthogonal reparameterization or simply from modality-specific parameterization.

4. Experiments on different modality pairs: The current evaluation focuses only on image-text multimodality. Given that the motivation emphasizes modality-specific discrepancies, extending experiments to text-audio tasks would better validate the method’s universality and its potential impact on broader multimodal fine-tuning.

5. Evaluation on different scale models: Although the authors acknowledge hardware constraints, all experiments are conducted on LLaVA-1.6-Mixtral-7B. It remains unclear whether the observed gains persist for larger backbones (e.g., 13B, 34B) or for architectures with stronger multimodal alignment such as Qwen-VL series or InternVL series. Evaluating scalability to larger models would substantiate the method’s general applicability.

**Questions:**

Please see weaknesses.

---

> ### Author Response · Authors · 2025-11-25
>
> Thank you for your valuable suggestions and your patience! We will address your concerns point by point. If you have any unresolved questions, please feel free to continue asking!
>
> ## 1. Explanation of Orthogonality (Weakness1)
>
> Thank you for pointing out the issues! It needs to be clarified that the orthogonality in our method exists between the parameter matrices rather than embeddings. This is explained in more detail in Section 3.1 of the main text. We will further emphasize this point and address any ambiguous phrasing.
>
> The data analysis results shown in Figure 3 of the paper provide evidence supporting this conclusion. After modality decoupling, the orthogonality of the column space of the parameter matrices is greatly enhanced (specifically, the value of $Decoupled\ U$ in the figure is much smaller than that of $Vanilla\ U$). Moreover, we performed the same orthogonality analysis on multiple datasets and modalities, and the results, as shown in our discussion with Reviewer Ngdi in Topic 2, confirm this finding.
>
> We believe this occurs because the text and visual embeddings differ in patterns, leading to some distinctions in the associated parameter matrices. Additionally, due to LoRA's low-rank nature, the parameter matrices are "thin" matrices, meaning one dimension is much larger than the other, making orthogonality more likely to occur.
>
> ## 2. Theoretical Explanation of Orthogonality and Generalization Ability, and Covariance (Weakness2)
>
> Orthogonality is the strongest form of independence. By maintaining orthogonality between modalities, the features encoded by each modality can eliminate redundant information, emphasizing the modality-specific information. This helps alleviate the issue of modality information confusion when embeddings of all modalities are processed similarly, thus reducing the covariance between modalities. This is further explained in Topic 4 of our discussion with Reviewer ALFh. For more detailed information, please refer to our response to this issue.
>
> We attempted further theoretical derivation. Suppose, during forward propagation, we have text and visual embeddings with batch size $b$ and dimension $d$, denoted as $T_t, T_v \in \mathbb{R}^{d \times b}$, and parameter matrices $A_t, A_v \in \mathbb{R}^{d^{'} \times d}$. During forward propagation, we have:
> $$
> T_t^{'}=A_tT_t
> $$
> $$
> T_v^{'}=A_vT_v
> $$
> To explore the redundancy of feature dimensions in text and visual embeddings, we can measure it by the covariance. Let $\overline{T_t}$ and $\overline{T_v}$ be the matrices consisting of the mean of the row vectors of $T_t$ and $T_v$, respectively. Then, we have:
> $$
> Cov(T_t, T_v) = \frac{1}{b-1}(T_t-\overline{T_t})(T_v-\overline{T_v})^T \in \mathbb{R}^{d\times d}
> $$
> After forward propagation, the covariance becomes:
> $$
> Cov(T_t^{'}, T_v^{'}) = \frac{1}{b-1}(A_tT_t-A_t\overline{T_t})(A_vT_v-A_v\overline{T_v})^T=A_tCov(T_t, T_v)A_v^T \in \mathbb{R}^{d^{'}\times d^{'}}
> $$
> Performing SVD on $Cov(T_t, T_v)$ gives:
> $$
> Cov(T_t^{'}, T_v^{'})=A_tU_c\Sigma_cV_cA_v^T
> $$
> where:
> $$
> U_c=[u^{(0)}, u^{(1)}, ...u^{(d-1)}],
> V_c=\left[\begin{matrix}v^{(0)} \\\\
> v^{(1)} \\\\
> \vdots \\\\
> v^{(d-1)}
> \end{matrix}\right],
> \Sigma_c=\begin{pmatrix}
> \sigma^{(0)} & 0 & \cdots & 0 \\\\
> 0 & \sigma^{(1)} & \cdots & 0 \\\\
> \vdots & \vdots & \ddots & \vdots \\\\
> 0 & 0 & \cdots & \sigma^{(d-1)} \\\\
> \end{pmatrix}
> $$
> Thus, $Cov(T_t, T_v)$ and $Cov(T_t^{'}, T_v^{'})$ can be expressed as:
> $$
> Cov(T_t, T_v)=[u^{(0)}, u^{(1)}, ..., u^{(d-1)}]
> \begin{pmatrix}
> \sigma^{(0)} & 0 & \cdots & 0 \\\\
> 0 & \sigma^{(1)} & \cdots & 0 \\\\
> \vdots & \vdots & \ddots & \vdots \\\\
> 0 & 0 & \cdots & \sigma^{(d-1)} \\\\
> \end{pmatrix}
> \left[\begin{matrix}v^{(0)} \\\\
> v^{(1)} \\\\
> \vdots \\\\
> v^{(d-1)}
> \end{matrix}\right]
> =\sum^{d-1}_{i=0}\sigma^{(i)}u^{(i)}v^{(i)}
> $$
>
> $$
> Cov(T_t^{'}, T_v^{'})=
> \left[\begin{matrix}a_{t}^{(0)} \\\\
> a_{t}^{(1)} \\\\
> \vdots \\\\
> a_{t}^{(d^{'}-1)} \\\\
> \end{matrix}\right]
> [u^{(0)}, u^{(1)}, ..., u^{(d-1)}]
> \begin{pmatrix}
> \sigma^{(0)} & 0 & \cdots & 0 \\\\
> 0 & \sigma^{(1)} & \cdots & 0 \\\\
> \vdots & \vdots & \ddots & \vdots \\\\
> 0 & 0 & \cdots & \sigma^{(d-1)} \\\\
> \end{pmatrix}
> \left[\begin{matrix}v^{(0)} \\\\
> v^{(1)} \\\\
> \vdots \\\\
> v^{(d-1)} \\\\
> \end{matrix}\right]
> [a_{v}^{(0)}, a_{v}^{(1)}, ...a_{v}^{(d^{'}-1)}]=
> \begin{pmatrix}
> c_{0,0}  & \cdots & c_{0,d^{'}-1} \\\\
> \vdots  & \ddots & \vdots \\\\
> c_{d^{'}-1,0} & \cdots & c_{d^{'}-1,d^{'}-1} \\\\
> \end{pmatrix}
> $$
>
> Therefore, the covariance between the $m$-th feature of text embeddings and the $n$-th feature of visual embeddings after forward propagation can be expressed as:
> $$
> c_{m,n}=a_{t}^{(m)}(\sum^{d-1}_{i=0}\sigma_{i}u_{i}v_{i})a_{v}^{(n)}=\sum^{d-1}_{i=0}a_{t}^{(m)}u_{i}\sigma_{i}v_{i}a_{v}^{(n)}
> $$
> where $a_{t}^{(m)}, u_{i}, v_{i}, a_{v}^{(n)}$ are unit vectors. In the case of $A_t=A_v$, $a_{t}^{(m)}=a_{v}^{(n)}$, and in the case of $A_t \perp A_v$, $a_{t}^{(m)} \perp a_{v}^{(n)}$, which makes the final value more robust.

---

> ### Author Response · Authors · 2025-11-25
>
> ## 3. Visual Evidence for Orthogonality Improving Multimodal Problems (Weakness3)
>
> Thank you for your valuable suggestion. We extracted the intermediate states of the forward propagation for the original LoRA and our method and calculated the correlation matrix. The resulting data is shown below. We will add the generated images to the final version. The table shows the correlation between the feature dimensions of text embeddings and visual embeddings. We selected a 4x4 subset of the data, which corresponds to the first four dimensions of text embeddings and visual embeddings.
>
> Correlation obtained with our method:
>
> | Column 1 | Column 2 | Column 3 | Column 4 |
> | :------: | :------: | :------: | :------: |
> |  0.0345  |  0.0309  |  0.0349  |  0.0309  |
> |  0.0296  |  0.0281  |  0.0329  |  0.0266  |
> |  0.0377  |  0.0257  |  0.0357  |  0.0278  |
> |  0.0298  |  0.0263  |  0.0256  |  0.0249  |
>
> Correlation obtained with LoRA:
>
> | Column 1 | Column 2 | Column 3 | Column 4 |
> | :------: | :------: | :------: | :------: |
> |  0.0411  |  0.0369  |  0.0324  |  0.0236  |
> |  0.0424  |  0.0352  |  0.0387  |  0.0306  |
> |  0.0395  |  0.0380  |  0.0447  |  0.0360  |
> |  0.0339  |  0.0406  |  0.0270  |  0.0374  |
>
> Difference between the two:
>
> | Column 1 | Column 2 | Column 3 | Column 4 |
> | :------: | :------: | :------: | :------: |
> | -0.0066  | -0.0060  |  0.0025  |  0.0072  |
> | -0.0127  | -0.0071  | -0.0058  | -0.0040  |
> | -0.0017  | -0.0122  | -0.0090  | -0.0082  |
> | -0.0041  | -0.0143  | -0.0014  | -0.0124  |
>
> As can be seen, our method significantly reduces the correlation between features of the two modalities, and the covariance shows a similar pattern. This demonstrates that the features extracted through orthogonal projection in our method can eliminate redundancy between modalities, mitigating the problem of modality information confusion. This is consistent with what we mentioned in Topic 4 of our discussion with Reviewer ALFh. More detailed explanations can be found in our response to this issue.
>
> ## 4. Experiment on Non-Orthogonalized Modality Decoupling Matrices (Weakness4)
>
> In Table 4 of the main text, the "Decoupled" row corresponds to the case where the image and text branches use different low-rank matrices trained independently and without orthogonalization. The difference with ORFLEX is shown, which reflects the effectiveness of our design.
>
> ## 5. Performance on More Modal Datasets (Weakness5)
>
> We attempted to evaluate our method on the audio modality, using Qwen2-Audio-7B-Instruct as the base model, and obtained the following results. It can be seen that our method also performs well on the audio modality. For the video modality, more experimental resources are needed. We are conducting further experiments, and we will report the results once available.
>
> |            | AdaLoRA | LoRA  | R-LoRA | VB-LoRA | VERA  | DoRA  | PiSSA | Ours  |
> | ---------- | ------- | ----- | ------ | ------- | ----- | ----- | ----- | ----- |
> | Clotho-AQA | 52.73   | 58.20 | 57.32  | 49.90   | 50.59 | 58.11 | 57.62 | 59.47 |
>
> ## 6. Performance on Larger Models (Weakness6)
>
> We attempted to validate our method on a larger model, InternVL3.5-14B, and obtained the following results. R-LoRA and DoRA triggered OOM (Out of Memory) errors, so we removed these two baseline methods. The results show that our method performs well on larger models. For models larger than 14B, we found that OOM errors occurred, and we were unable to further validate them.
>
> |         | A-OKVQA | OCR-VQA | TextVQA | AVERAGE-ACC |
> | :-----: | :-----: | :-----: | :-----: | :---------: |
> | AdaLoRA |  86.91  |  82.42  |  68.85  |    79.39    |
> |  LoRA   |  87.70  |  84.96  |  69.33  |    80.66    |
> | VB-LoRA |  86.13  |  84.86  |  68.95  |    79.98    |
> |  VERA   |  87.40  |  82.03  |  42.48  |    70.64    |
> |  PiSSA  |  87.79  |  84.96  |  68.94  |    80.56    |
> |  Ours   |  88.18  |  85.64  |  69.63  |    81.09    |

---

### Official Review · Reviewer_Ngdi · 2025-10-31

**Soundness:** 2
**Presentation:** 2
**Contribution:** 2
**Rating:** 4
**Confidence:** 2

**Summary:**

This work observes that visual and text tokens in MLLMs, though projected into the same feature space, still exhibit significant distribution differences during forward propagation. The authors find that LoRA updates for the two modalities tend to become orthogonal when decoupled. Motivated by this, they propose ORFLEX, a modality-aware PEFT framework that introduces separate reparameterization branches for visual and text tokens and decomposes LoRA matrices into a frozen orthogonal basis and a learnable component.

**Strengths:**

1,Modality-aware parameter design: Identifies and leverages intrinsic differences between visual/text token spaces, addressing a real gap in previous LoRA methods.

2,Clear theoretical grounding: Uses linear algebra tools (QR decomposition, Householder transforms) to enforce orthogonality with mathematical rigor.

3,Strong empirical results: Demonstrates consistent gains across general and domain-specific MLLM benchmarks.

**Weaknesses:**

1, Extra complexity / parameters: Dual branches and orthogonality maintenance introduce additional overhead compared to standard LoRA.

2, Orthogonality assumption universality: Strong orthogonality observation may depend on specific architectures or datasets; unclear if it holds broadly.

3, Limited modality scaling discussion: Method focuses on two modalities; generalization to audio/video/3D or >2 modalities is not explored.

**Questions:**

1, Layer-wise behavior:

Does visual-text token orthogonality vary across model depth? Would applying decoupling only at selective layers achieve similar performance with lower cost?

2, Scalability beyond two modalities:

How would this method generalize if more modalities (e.g., audio, depth, video tokens) are introduced, where pairwise orthogonality may not hold cleanly?

---

> ### Author Response · Authors · 2025-11-25
>
> Thank you for your valuable suggestions and your patience! We will address your concerns point by point. If you have any unresolved questions, please feel free to continue asking!
>
> ## 1. Comparison of Costs between Our Method and Baselines (Weakness1)
>
> We conducted tests using InternVL3.5-8B on the AOKVQA dataset to quantitatively demonstrate the cost comparison among various baselines. The specific data is as follows. While our method does introduce more complexity, the additional cost is acceptable when compared to the baselines, as they exhibit higher memory and time costs. Although our method appears to have several modules, the number of trainable parameters has not increased; we only introduce some frozen parameters and Householder transformations, which increase the FLOPs. The impact on time/memory cost is acceptable.
>
> |            | AdaLoRA | LoRA  | R-LoRA | VB-LoRA | VERA  | DoRA  | PiSSA | Ours  |
> | :--------: | :-----: | :---: | :----: | :-----: | :---: | :---: | :---: | :---: |
> | GPU Memory |  40.81  | 40.73 | 43.60  |  40.82  | 41.16 | 43.63 | 40.74 | 43.57 |
> | time/step  |  5.85   | 5.75  |  5.75  |  6.06   | 5.71  | 5.24  | 5.49  | 5.99  |
> |   flops    |  59.47  | 59.61 | 59.68  |  59.61  | 59.61 | 59.79 | 59.61 | 59.99 |
>
> ## 2. Is Strong Orthogonality Specific to Certain Datasets or Modalities? (Weakness2&3, Problem2)
>
> To verify this issue, we performed orthogonality analysis on multiple datasets and modalities (text-image, text-audio), similar to the analysis in the main text. The results are as follows. It can be seen that orthogonality of parameter matrix subspaces still exists significantly across modalities and datasets. Furthermore, for fusion problems involving more than two modalities, we are conducting additional experiments, and we will report the results once they are obtained.
>
> We believe this phenomenon occurs because, in multimodal problems, different modalities generally contain unique information that other modalities do not have, which is a common feature. This leads to differences in the parameter matrices associated with each modality. Additionally, due to the low-rank nature of LoRA, one dimension of the parameter matrix is much larger than the other, making orthogonality more likely to occur.
>
> |                 | A-OKVQA | OCR-VQA | ScienceQA | TextVQA | Clotho-AQA |
> | :-------------: | :-----: | :-----: | :-------: | :-----: | :--------: |
> |  $Q_{vanilla}$  |  0.220  |  0.193  |   0.158   |  0.136  |   0.176    |
> | $Q_{decoupled}$ |  0.056  |  0.054  |   0.057   |  0.055  |   0.058    |
> |  $K_{vanilla}$  |  0.216  |  0.189  |   0.162   |  0.137  |   0.178    |
> | $K_{decoupled}$ |  0.056  |  0.054  |   0.056   |  0.056  |   0.059    |
> |  $V_{vanilla}$  |  0.161  |  0.165  |   0.143   |  0.137  |   0.153    |
> | $V_{decoupled}$ |  0.054  |  0.055  |   0.056   |  0.055  |   0.057    |
> |  $O_{vanilla}$  |  0.164  |  0.162  |   0.145   |  0.135  |   0.149    |
> | $O_{decoupled}$ |  0.093  |  0.078  |   0.081   |  0.084  |   0.080    |
>
> ## 3. Does Visual-Textual Orthogonality Change with Model Depth? (Problem1)
>
> First, it should be clarified that in the paper, orthogonality is always referred to the parameter matrix subspace, rather than the embeddings. This is explained in more detail in section 3.1 of the main text, and we will further emphasize this point and address any ambiguous phrasing that could lead to misunderstanding.
>
> Regarding the orthogonality of subspaces in different layers of the model, our experimental analysis concludes that orthogonality exists across all layers. Figure 3 in the main text supports this, where the boxplot's data distribution is derived from the analysis results of layers 1-32 of the model. As shown, under the decoupled condition, the boxes tend to be very compact, indicating that the variance of orthogonality within each layer is small, and therefore, high orthogonality exists within each layer.
>
> Additionally, using the A-OKVQA dataset as an example, we present detailed orthogonality data for each layer, which further supports this conclusion:
>
> |      | $Layer\ 0$ | $Layer\ 8$ | $Layer\ 16$ | $Layer\ 24$ |
> | :--: | :--------: | :--------: | :---------: | :---------: |
> | $Q$  |   0.050    |   0.056    |    0.058    |    0.053    |
> | $K$  |   0.055    |   0.053    |    0.053    |    0.054    |
> | $V$  |   0.053    |   0.054    |    0.051    |    0.051    |
>
> In fact, we believe that our orthogonality design is not just tailored to the orthogonality properties of decoupled LoRA, but also plays an important role in multimodal problems by removing the extra redundancy between different feature dimensions of the embeddings, which helps mitigate the issue of modality information confusion. This is important throughout the entire forward propagation process of the model. For more detailed discussions, please refer to Topic 4 of our conversation with Reviewer ALFh.

---

### Note · Authors · 2026-01-27

I have read and agree with the venue's withdrawal policy on behalf of myself and my co-authors.

---

### Meta-Review · Area_Chair_5JHC · 2026-01-07

**Summary:**

Reviewers generally regard the paper as a solid and well-presented contribution that introduces a modality-aware LoRA design grounded in rigorous linear algebra, with consistent empirical gains across multiple multimodal benchmarks. The orthogonality-based reparameterization is viewed as technically sound and parameter-efficient, and the ablation studies are appreciated for supporting the design choices. However, several concerns are repeatedly raised. The motivation and interpretation of “orthogonality” remain partially unclear, particularly how parameter-space orthogonality relates to semantic alignment in multimodal representations. The novelty and necessity relative to existing PEFT methods (e.g., DoRA, PiSSA, full fine-tuning) are not fully convincing, given that performance gains over LoRA are sometimes modest. Reviewers also highlight limited generalizability, as experiments focus only on image–text settings and a single mid-sized backbone, with no validation on larger models or additional modalities. Finally, the method introduces extra complexity, yet training time, memory overhead, and scalability costs are not quantified, leaving practical efficiency claims insufficiently supported. Despite the authors' efforts during the rebuttal, they did not receive sufficient support. Therefore, AC's recommendation is to reject.

**Reviewer Concerns:**

The rebuttal clarified the concept of parameter space orthogonality and provided a detailed theoretical proof. The authors supplemented the experiment to demonstrate the applicability of the proposed method to larger backbone networks and architectures with stronger multimodal alignment. The authors claim that their use of fewer trainable parameters demonstrates an efficiency advantage, but more in-depth experiments did not support this assertion.

**Reviewer Scores:**

I expect the final rating to be as follows:
- Reviewer Ngdi: 4
- Reviewer rEFZ: 4
- Reviewer ALFh: 4
- Reviewer mUpE: 4

---

### Decision · Program_Chairs · 2026-01-26

Reject